# COUNT: COntrastive UNlikelihood Text Style Transfer for Text Detoxification

**Mohammad Mahdi Abdollah Pour[1], Parsa Farinneya[1],**
**Manasa Bharadwaj[2], Nikhil Verma[2], Ali Pesaranghader[2]** and **Scott Sanner[1]**

[1] University of Toronto, Canada

{m.abdollahpour,parsa.farinneya}@mail.utoronto.ca, ssanner@mie.utoronto.ca

[2] LG Electronics, Toronto AI Lab

{manasa.bharadwaj,nikhil.verma,ali.pesaranghader}@lge.com

## Abstract

Offensive and toxic text on social media platforms can lead to polarization and divisiveness within online communities and hinders constructive dialogue. Text detoxification is a crucial task in natural language processing to ensure the generation of non-toxic and safe text. Text detoxification is a special case of the Text Style Transfer (TST) problem, where an input text is rephrased to an output text that preserves its content while modifying the style (in this case to a more neutral, non-toxic style). State-of-the-art methods for detoxification use supervised training of encoder-decoder models to produce gold-standard outputs with a standard likelihood-based objective. However, it can be hard for these models to deviate from their pretrained auto-encoder identity mapping. While previous methods have used unlikelihood-based losses to penalize input-to-output copying of toxic content, these methods also unfortunately penalize non-toxic content in the input that would be fine to preserve in the output. To address these issues, we introduce a novel contrastive unlikelihood objective (COUNT[1]) that directly contrasts the gold standard rephrasing with the identity input-to-output mapping to effectively isolate and focus learning on non-toxic style transfer. We benchmark COUNT on two parallel datasets, ParaDetox and APPDIA, showing that it achieves significant improvements in jointly combined fluency, content preservation, and detoxification (i.e., the highest "J" score).

## 1 Introduction

***Disclaimer.*** *Please be aware that as you read this paper, you may come across texts that could be considered toxic due to the nature of this research.*

Exposure to offensive and toxic text on online platforms can have detrimental consequences, causing emotional distress and negative psychological effects on users (González-Bailón and Lelkes, 2023). Such experiences can deter individuals from actively engaging in online communities and social media. Therefore, text detoxification solutions are indispensable for mitigating harmful content on online platforms.

Text detoxification is a text-style transfer task where the goal is to change the style of an input text from toxic to safe while the content is preserved. Most state-of-the-art text detoxification methods use supervised training of encoder-decoder models (Logacheva et al., 2022). The objective of those methods relies on maximizing the likelihood of generating gold-standard non-toxic outputs, but fails to penalize generation of the toxic inputs. To address this issue, Welleck et al. (2019) proposed an "unlikelihood" training methodology that can be used to penalize generation of toxic inputs while encouraging generation of gold standard non-toxic output. While this improves on the simple likelihood objective, we observe experimentally that it often fails to overcome the strong inertia of the pretrained model to produce the identity input-to-output mapping that preserves toxic text. One key reason for this is that unlikelihood cannot be weighted too heavily can penalize both the toxic and non-toxic content in the input. To address this issue, we introduce a novel **C**ontrastive **Un**likelihood **T**ext style transfer (COUNT) objective that directly contrasts the likelihood of the gold standard rephrasing with the likelihood of the identity input-to-output mapping to effectively isolate and focus learning on style transfer for text detoxification.

Our contributions are twofold: (1) We propose the COUNT loss function as a novel training objective for text detoxification. (2) We show that by using the COUNT loss, our method delivers significant improvements on a joint measure ("J" score) of fluency, content preservation, and detoxification on two publicly available parallel datasets, ParaDetox and APPDIA, for the text detoxification task.

---

[1] https://github.com/D3Mlab/count-style-transfer

## 2 Related Work

### 2.1 Text Detoxification

Text detoxification solutions primarily fall into two main categories, *unsupervised* and *supervised*. We describe each category below:

**Unsupervised.** The unsupervised methods are typically built on a non-parallel dataset, which is a set of toxic and a set of non-toxic texts without one-to-one mappings between them. Mask&Infill (Wu et al., 2019), DRG-Template/Retrieve (Li et al., 2018), CondBERT (Dale et al., 2021) and SST (Lee, 2020) perform pointwise correction of the tokens that mostly affect the style. DLSM (He et al., 2020) and Luo et al. (2019) train the encoder-decoder detoxification model using non-parallel data. He et al. (2020) uses amortized variational inference. Luo et al. (2019) use reinforcement learning and a style classifier. CAE-T5 (Laugier et al., 2021) tried to balance content preservation and style accuracy in an unsupervised setting using a loss made of conditional auto-encoding and cycle consistency. Shen et al. (2017) and Fu et al. (2018) use adversarial classifiers to adapt their decoder to generate text of non-toxic style. ParaGeDi (Dale et al., 2021) uses a paraphraser model guided by style-trained language models for detoxification. In this work, we focus on supervised methods discussed next that typically perform better.

**Supervised.** In contrast, the supervised methods are usually built on parallel datasets in which one-to-one mappings between toxic and non-toxic texts exist. ParaDetox (Logacheva et al., 2022) fine-tunes BART (Lewis et al., 2020) on their parallel data. APPDIA (Atwell et al., 2022) introduces a discourse-aware style-transfer model to reduce offensiveness in the text while preserving the semantics of the original text. While Logacheva et al. (2022) and Atwell et al. (2022) achieve some of the most promising results for detoxification, their methods do not fully penalize the toxic style in the training process and thus fail to detoxify inputs in many cases as we demonstrate empirically. This is the aspect we aim to improve in this work.

### 2.2 Penalizing Negative Examples

Previous work on detoxification did not systematically address penalizing the toxic samples. For instance, ParaGeDi (Dale et al., 2021) proposes a loss to penalize the generated texts that are labeled as toxic by toxicity classifiers. However, the multi-ple objective nature of text style transfer (preserve content, preserve fluency, change style) has not been properly and systematically addressed as ParaGeDi relies on heuristics for content preservation.

Unlikelihood training has been used in controllable text generation applications to avoid undesirable tokens with a high probability (Welleck et al., 2019). CLICK (Zheng et al., 2023), SLiC (Zhao et al., 2022), BRIO (Liu et al., 2022) and CRINGE (Adolphs et al., 2022) also use unlikelihood training for various text generation applications such as summarization and sentiment control. Despite the popularity of unlikelihood training in text generation, it has not been widely applied to the text style transfer task. We empirically observe that unlikelihood (Welleck et al., 2019) training for text style transfer often fails to properly penalize the toxic samples since it has two training objectives that often conflict. Specifically, penalizing the likelihood of generating the toxic sample contradicts with maximizing the likelihood of the gold-standard target since the gold-standard rephrasing often overlaps substantially with the input and the loss cannot discriminate the toxic from the non-toxic content. Addressing this deficiency of unlikelihood for detoxification is our primary focus in this work.

## 3 Methodology

The text-detoxification task aims to generate a non-toxic sentence $y$ given a toxic input $x$ while preserving the content of $x$.

State-of-the-art detoxification methods (Logacheva et al., 2022) fine-tune an encoder-decoder model such as BART (Lewis et al., 2020) for detoxification. BART is a sequence-to-sequence denoising autoencoder that has been trained to reconstruct the input text. The standard approach for fine-tuning encoder-decoder models maximizes the probability of generating target sequence $y^*$ by the decoder, given the input $x$ to the encoder. This is done using the standard Language Modeling (LM) loss:

$$\mathcal{L}_{LM} = \mathbb{E}_{(x,y^*)\sim\mathcal{D}}[-\log(p_\theta(y = y^*|x))] \quad (1)$$

where $y^*$ is a non-toxic paraphrase of toxic text $x$. The $(x, y^*)$ pairs are available in the parallel detoxification dataset $\mathcal{D}$. $p_\theta(y = y^*|x)$ is the likelihood of string $y^*$ as the output when $x$ is given as the input to the encoder-decoder model parametrized by $\theta$.

For fine-tuning, Eq. (1) encourages the output to have a non-toxic style by training the model to have a high likelihood for $y^*$ (learning from parallel data). However, in the detoxification task, we need to also penalize the toxicity in addition to maximizing the likelihood of target sequence $y^*$. Since encoder-decoder models such as BART reconstruct the input, these models need to deviate from the identity mapping to reduce toxicity.

Unlikelihood training (Welleck et al., 2019) can be used to penalize toxicity by minimizing the likelihood of toxic samples using the following Unlikelihood Training (UT) loss (Welleck et al., 2019):

$$\mathcal{L}_{UT} = \mathbb{E}_{x \sim \mathcal{D}}[-\log(1 - p_\theta(y = x|x))] \quad (2)$$

Then we would optimize the total loss

$$\mathcal{L}_{total-UT} = \mathcal{L}_{LM} + \alpha \cdot \mathcal{L}_{UT}. \quad (3)$$

However, minimizing the likelihood of $p_\theta(y = x|x)$ partially interferes with maximizing the likelihood of $p_\theta(y = y^*|x)$ in $\mathcal{L}_{LM}$ since $x$ and $y^*$ have similar content in the detoxification task and only differ in their style (toxic vs. non-toxic).

Therefore, we propose the COUNT loss function to discriminatively focus on detoxifying the style of the text. Specifically, the COUNT loss function contrasts toxic sentence $x$ and target non-toxic paraphrase $y^*$ as they have the same semantics but a different style. Hence, contrastively maximizing the likelihood of $p_\theta(y = y^*|x)$ relative to the likelihood of $p_\theta(y = x|x)$ helps normalize for properties such as text length and the likelihood of shared words between $x$ and $y^*$ that are irrelevant to style while focusing on maximizing the gap between them (i.e., the toxic component). Formally, the COUNT loss function is defined as follows:

$$\mathcal{L}_{COUNT} = \mathbb{E}_{(x,y^*) \sim \mathcal{D}}$$
$$\left[ -\log\left( \frac{p_\theta(y = y^*|x)}{p_\theta(y = y^*|x) + p_\theta(y = x|x)} \right) \right] \quad (4)$$

To minimize the loss in Eq. (4), the model is aiming for a large probability of the non-toxic reference $p_\theta(y = y^*|x)$ relative to the identity decoding $p_\theta(y = x|x)$. However, as we show in the next section, this loss aims to avoid penalizing the common (non-toxic) content between $y^*$ and $y$ since the probability of this content cancels in the ratio.

We combine the COUNT loss with the standard language modeling loss so the total loss would be:

$$\mathcal{L}_{total-COUNT} = \mathcal{L}_{LM} + \alpha \cdot \mathcal{L}_{COUNT} \quad (5)$$

$\alpha$ is a hyperparameter and the value for $\alpha$ is chosen on validation data (cf. Appendix A for details).

## 3.1 Comparative analysis of COUNT vs UT

We now compare and contrast the COUNT loss to the UT loss to understand their differences. Formally, in the UT loss of Eq. (2) and the COUNT loss of Eq. (4), let us assume we can factorize the likelihood of a target style transfer example into two components such that $p_\theta(y = y^*|x) = p_c^x \cdot p_{nt}^{y^*}$, where $p_c^x$ is the probability of common content between the input $x$ and the target $y^*$ and $p_{nt}^{y^*}$ is the probability of non-toxic content in $y^*$. Similarly, let us assume that the likelihood of the identity decoding factorizes as $p_\theta(y = x|x) = p_c^x \cdot p_t^x$, where $p_c^x$ is the same as before and $p_t^x$ is the probability of the toxic content in $x$. Then we can respectively rewrite our two losses in Eq. (2) and (4) as follows:

$$\mathcal{L}_{UT} = \mathbb{E}_{x \sim \mathcal{D}}[-\log(1 - p_c^x \cdot p_t^x)] \quad (6)$$

$$\mathcal{L}_{COUNT} = \mathbb{E}_{(x,y^*) \sim \mathcal{D}} \left[ -\log \frac{p_c^x \cdot p_{nt}^{y^*}}{p_c^x \cdot p_{nt}^{y^*} + p_c^x \cdot p_t^x} \right]$$

$$= \mathbb{E}_{(x,y^*) \sim \mathcal{D}} \left[ -\log \frac{p_c^x \cdot p_{nt}^{y^*}}{p_c^x \cdot (p_{nt}^{y^*} + p_t^x)} \right]$$

$$= \mathbb{E}_{(x,y^*) \sim \mathcal{D}} \left[ -\log \frac{p_{nt}^{y^*}}{p_{nt}^{y^*} + p_t^x} \right] \quad (7)$$

In both cases, we want to minimize the overall loss, which is equivalent to maximizing the quantity inside the $-\log[\cdot]$. Here we can see that the UT loss aims to maximize $1 - p_c^x \cdot p_t^x$, which is equivalent to minimizing $p_c^x \cdot p_t^x$; hence, it aims to minimize the joint probability of *both* the common and toxic components. In contrast, the COUNT loss *cancels* any contribution of the common components $p_c^x$ and is maximized as the probability of the non-toxic content $p_{nt}^{y^*}$ increases relative to the toxic content $p_t^x$. In this way, the COUNT loss is able to isolate and minimize toxic content while the UT loss additionally penalizes common non-toxic content between $y^*$ and $x$.

As a final remark for completeness, both UT and COUNT provide a mixture of their loss with the LM loss, which is not considered in the analysis above, but which can be considered an orthogonal loss simply intended to ensure the output achieves high likelihood under a standard language model.

In terms of time complexity, $\mathcal{L}_{LM}$ requires computing $p_\theta(.|.)$ only once for $p_\theta(y = y^*|x)$. $\mathcal{L}_{UT}$ requires computing $p_\theta(y = x|x)$ (i.e., $y$ is an

identity copy of $x$), therefore the full $\mathcal{L}_{total-UT}$ (including the LM component) requires computing $p_\theta(.|.)$ twice. $\mathcal{L}_{COUNT}$ requires computing $p_\theta(y = y^*|x)$ and $p_\theta(y = x|x)$, therefore the full $\mathcal{L}_{total-COUNT}$ also requires computing $p_\theta(.|.)$ twice since COUNT's $p_\theta(y = y^*|x)$ computation can be reused for the LM component. Hence, the final COUNT and UT losses each have the same computational cost, which is a constant two multiple of the computational cost of the LM loss.

## 4 Experiments

### 4.1 Experimental Settings

**Datasets.** We study two parallel datasets for evaluating detoxification methods: **ParaDetox** (Logacheva et al., 2022) contains non-toxic paraphrases of toxic sentences. The partitioning of the dataset includes two sets, one for training with 11,939 sentence pairs and the other for testing with 671 sentences rephrases. **APPDIA** (Atwell et al., 2022) was created using comments collected from the well-known social discussion website Reddit. The authors collected nearly 2,000 toxic comments which were then annotated by experts in the field of sociolinguistics to generate the non-toxic version of toxic sentences. The dataset has an 80-10-10 split for train-validation-test splitting.

**Metrics.** We follow the well-established text detoxification work (Logacheva et al., 2022) to evaluate our experiments with **BLEU**, Style Accuracy (**STA**), Semantic Similarity or Content Preservation (**SIM**), Fluency (**FL**), and **J** score. In particular, STA and FL are computed with pre-trained classifiers (Warstadt et al., 2019) to measure the non-toxicity and fluency of a given sentence, respectively. SIM is computed using cosine similarity between the input and the generated detoxified text with the model of Wieting et al. (2019). Moreover, we compute J score (Krishna et al., 2020) as the average product of STA, SIM, and FL.

### 4.2 Implementation Details

We use an Adam optimizer (Kingma and Ba, 2015) with a learning rate of 5e-5 and use a validation set of 20% for Paradetox hyperparameter tuning. The validation set for APPDIA is provided by the dataset. The evaluation toolkit is the same as Logacheva et al. (2022)[2].

[2]https://github.com/s-nlp/paradetox#detoxification-evaluation

### 4.3 Experimental Results

Tables 1 and 2 show the performance of the proposed COUNT loss (Eq. (5)) function compared to the original LM loss function (Eq. (1)) and the UT loss (Eq. (3)) for training BART on the Paradetox and APPDIA datasets. The COUNT loss function provides significant improvement in both datasets over the original LM loss function and UT loss. While SIM and FL are almost similar in original loss function and the proposed loss in Paradetox, the improvement in the J score is due to the improvement in STA. Similarly for APPDIA, while FL and SIM are lower, the improvement in J comes from a significant improvement in STA. The improvements in the STA scores show that the proposed COUNT loss function focuses on penalizing the toxic style which leads to higher STA scores and a better detoxification performance in general. Detoxified examples can be found in Appendix B.

Lower SIM scores for COUNT loss are reasonable since the examples in the APPDIA dataset need a more substantial rewriting to be detoxified compared to Paradetox. Hence, as the paraphrased texts improve they tend to be less similar to the original toxic text that was used as input (due to deletion or modification of the toxic part).

Based on results in Tables 1 and 2, the UT loss performs better than the original LM loss in STA score but it does not do as well as the COUNT loss in penalizing the toxic style and thus yields lower STA scores compared to the COUNT loss and lower J scores. The reason is that penalizing the toxicity in UT loss is in contrast with maximizing the likelihood of the gold standard target.

Table 1 compares the proposed method's performance to the Paradetox dataset's baselines. Results show that by using the proposed COUNT loss function, we can significantly improve the style transfer accuracy and outperform all of the baselines in the J score. DRG-Retrieve, CondBERT, ParaGeDi, and DiffuDetox (Floto et al., 2023) have better STA scores than the COUNT method but the mentioned baselines fail to have balanced performance among all of the three evaluation metrics (STA, SIM, FL) and mostly fail significantly in one of them causing them to have a lower J score. In contrast, our COUNT methodology exhibits a clear balance in performance among all task objectives.

Table 2 compares the proposed COUNT method with the baseline on the APPDIA dataset. BART, DialoGPT, and T5 are baselines from Atwell et al.

| | BLEU | STA | SIM | FL | J |
|---|---|---|---|---|---|
| Human | 100.0 | 0.96 | 0.77 | 0.88 | 0.66 |
| DRG-Template | 53.86 | 0.90 | 0.82 | 0.69 | 0.51 |
| DRG-Retrieve | 4.74 | 0.97 | 0.36 | 0.86 | 0.31 |
| Mask&Infill | 52.47 | 0.91 | 0.82 | 0.63 | 0.48 |
| CondBERT | 42.45 | 0.98 | 0.77 | 0.88 | 0.62 |
| SST | 30.20 | 0.86 | 0.57 | 0.19 | 0.10 |
| ParaGeDi | 25.39 | **0.99** | 0.71 | 0.88 | 0.62 |
| DLSM | 21.13 | 0.76 | 0.76 | 0.52 | 0.25 |
| DiffuDetox | 62.13 | 0.92 | 0.88 | 0.80 | 0.67 |
| ParaDetox | 64.53 | 0.89 | 0.86 | 0.89 | 0.68 |
| Original LM Loss | 71.84 | 0.89 | **0.89** | 0.90 | 0.71 |
| UT Loss | **72.43** | 0.87 | **0.89** | 0.90 | 0.71 |
| **COUNT (Ours)** | 69.68 | 0.91 | 0.88 | **0.91** | **0.74** |

Table 1: Text detoxification performance on the ParaDetox dataset. Baseline results are taken from (Logacheva et al., 2022). The best results are in boldface.

| | BLEU | STA | SIM | FL | J |
|---|---|---|---|---|---|
| Human | 60.18 | 0.87 | 0.77 | 0.95 | 0.65 |
| BART | 75.85 | 0.72 | 0.88 | 0.96 | 0.61 |
| DialoGPT | 45.12 | **0.85** | 0.70 | 0.81 | 0.46 |
| PDTB+RST | 49.46 | **0.85** | 0.73 | 0.87 | 0.53 |
| T5 | 74.69 | 0.82 | 0.88 | **0.97** | **0.70** |
| Original LM Loss | **79.00** | 0.69 | **0.91** | 0.97 | 0.60 |
| UT Loss | 72.65 | 0.80 | 0.88 | 0.95 | 0.60 |
| **COUNT (Ours)** | 68.99 | **0.85** | 0.85 | 0.93 | 0.68 |

Table 2: Text detoxification performance on the APP-DIA dataset. The best results are in boldface.

(2022), and PDTB+RST is the methodology proposed by Atwell et al. (2022). The APPDIA dataset is small and due to the stochasticity of the results on APPDIA dataset, we report the results as the average of 5 runs. Our COUNT methodology again achieves significant improvements in the STA score while having a comparable performance FL.

### 4.4 Failure Analysis and Future Directions

**Failure Analysis:** [**WARNING**: offensive content from an actual example.] In Appendix B, we present examples of the original input, gold standard (human) target, and outputs from training with the standard LM loss and COUNT loss. Overall, we remark that COUNT generally does an excellent job of removing explicit toxic content, but sometimes only by creating semantic ambiguity. Notably in the last example, COUNT has replaced "those tits" with "they", which has masked an explicitly toxic phrase with an ambiguous reference. We view this as a subtle semantic failure that is not likely to be caught by existing automated evaluation metrics and perhaps even some human evaluators.

**Directions for Future Improvements:** Given this previously observed failure mode of superficially masking toxicity via semantic ambiguity (which occurs not only with the COUNT loss, but with other methods), we remark that not only is it a failure, but it is often missed by evaluation metrics. This leads to two related but very different directions for future work: (i) It would be useful to investigate whether fine-tuning a stronger LM capable of deeper semantic reasoning (e.g., GPT-3.5 Turbo) may be able to penalize the likelihood of outputs with such semantic ambiguity under the COUNT loss, thus leading to better performance. (ii) Since we've also noted that existing evaluation metrics may fail to detect such cases in the first place, this also points to the need for better-automated toxicity evaluation, which may be remedied by investigating whether prompting methodologies with SOTA LMs like GPT-3.5/4 can better reason about these semantically subtle failure cases in ways that existing evaluation metrics fail to detect.

**Evaluation of Detoxification by Difficulty Level:** We have observed that detoxification difficulty can vary widely between test cases. To better understand comparative performance differences, we advocate that future work subdivide toxicity test cases and results analysis according to the amount of modification needed. For example, difficulty can range across the following categories of increasing hardness: (i) simple toxic word replacement, (ii) phrase replacement starting at the toxic word, (iii) phrase replacement starting before the toxic word (i.e., harder for most forward-pass beam search decoding methods), and (iv) substantial rephrasing to detoxify while preserving semantic content.

## 5 Conclusion

In this paper, we observed that existing methodologies for detoxification either fail to penalize input-to-output copying of toxic content (LM loss) or also penalize input-to-output copying of non-toxic content (UT loss). To resolve this, we introduced the novel COUNT loss function to contrastively penalize toxic text style in the detoxification task while effectively normalizing for other style-irrelevant aspects during training. Experimental results show the proposed COUNT method achieves significant improvements in the combined J-score metric for text detoxification on ParaDetox and APPDIA.

In general, the COUNT methodology is not specific to detoxification methods for text style transfer and has potential application to many other style transfer tasks that could be explored in future work.

## Limitations

A first limitation of this work is that it relies on parallel datasets for training which may not be easy to collect. A second limitation is that current detoxification datasets (Logacheva et al., 2022; Atwell et al., 2022) only focus on types of toxicities that use vulgar language. However, there are other types of toxicity that could be present in social media that do not necessarily use profanity or vulgar language. For example, cases of sarcasm, stereotypes, mockery, micro-aggression, etc. (Bhat et al., 2021). To leverage our methodology, there is a need for parallel detoxification datasets for a broader range of toxicity types.

## Ethical Considerations

**Potential Misuse:** Our method can be reversed and hypothetically used to obtain toxic sentences from non-toxic sentences. However, there are likely simpler ways for introducing toxicity that may mitigate the likelihood of this case for misuse.

**Environmental Cost:** We note that our work required extensive computational experiments to draw sound conclusions. However, models in production may not require such exhaustive experimentation and they can also be trained once using the most promising settings, thus mitigating future computational costs of this methodology.

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

Paradetox

| $\alpha$ | 0.2 | 0.5 | 1.0 | 2.0 | 3.0 | 4.0 | 5.0 |
|---|---|---|---|---|---|---|---|
| J | 0.694 | **0.710** | 0.700 | 0.703 | 0.700 | 0.706 | 0.701 |

APPDIA

| $\alpha$ | 0.2 | 0.5 | 1.0 | 2.0 | 3.0 | 4.0 | 0.5 |
|---|---|---|---|---|---|---|---|
| J | 0.652 | **0.700** | 0.670 | 0.671 | 0.622 | 0.597 | 0.637 |

Table 3: J score performance for different values of $\alpha$ for validation set in COUNT total loss

Paradetox

| $\alpha$ | 0.2 | 0.5 | 1.0 | 2.0 | 3.0 | 4.0 | 5.0 |
|---|---|---|---|---|---|---|---|
| J | 0.699 | 0.682 | 0.685 | 0.695 | 0.687 | 0.697 | **0.701** |

APPDIA

| $\alpha$ | 0.2 | 0.5 | 1.0 | 2.0 | 3.0 | 4.0 | 5.0 |
|---|---|---|---|---|---|---|---|
| J | 0.625 | 0.620 | 0.629 | **0.642** | 0.622 | 0.586 | 0.527 |

Table 4: J score performance for different values of $\alpha$ for validation set in UT total loss.

## A Appendix A: Hyperparameter selection

We choose the value of hyperparameter $\alpha$ in Eq. (5) and Eq. (3) based on J score performance on validation data over the set of values for $\alpha$. Tables 3 and 4 show the best value of $\alpha$ based on validation data performance along with performance for other values of $\alpha$ for all experiment scenarios.

## B Appendix B: Examples

Table 5 shows some examples of toxic sentences along with human reference paraphrase, prediction of LM loss method, and prediction of the COUNT method.

| Toxic | Dump his ass. Tablemanners are no rocket science. |
|---|---|
| Human | You should leave him. Tablemanners are no rocket science. |
| Standard Language Modeling Loss | Dump his head. Tablemanners are no rocket science. |
| COUNT Loss | Don't talk about it. Tablemanners are no rocket science. |
| Toxic | what's to investigate, his duplicitous incompetence is manifest |
| Human | What is left to investigate? |
| Standard Language Modeling Loss | what's to investigate, his duplicitous incompetence is manifest. |
| COUNT Loss | what's left to investigate? |
| Toxic | i m australian myself , but i don t say mate or cunt all the time . |
| Human | I am an Australian, but I don't say profanity words all the time. |
| Standard Language Modeling Loss | i m australian myself, but i don t say mate or cunt all the time. |
| COUNT Loss | i m australian myself, but i don t say mate or other things all the time. |
| Toxic | So it's not cheating if you fuck somebody and it never comes up? |
| Human | So it's not cheating if you sleep with somebody and it never comes up? |
| Standard Language Modeling Loss | So it's not cheating if you fuck somebody and it never comes up? |
| COUNT Loss | So it's not cheating if you don't talk to somebody and it never comes up? |
| Toxic | This is so cringe. What a fucking loser. |
| Human | This is so cringe. |
| Standard Language Modeling Loss | This is so cringe. What a loser. |
| COUNT Loss | This is so cringe. What a person. |
| Toxic | Hope the stupid cunt who did this to you gets shot |
| Human | Hope who did this to you pays |
| Standard Language Modeling Loss | Hope the person who did this to you gets shot |
| COUNT Loss | Hope the person who did this to you gets caught |
| Toxic | thank you, for acknowledging your own stupidity! |
| Human | thank you, for acknowledging the futility of your argument |
| Standard Language Modeling Loss | thank you, for acknowledging your own stupidity! |
| COUNT Loss | thank you, for acknowledging your own misinformedness |
| Toxic | I stopped listening to her when she first spoke but those tits keep me entertained. |
| Human | I stopped listening to her when she first spoke but her looks keep me entertained. |
| Standard Language Modeling Loss | I stopped listening to her when she first spoke but those tits keep me entertained. |
| COUNT Loss | I stopped listening to her when she first spoke but they keep me entertained. |

Table 5: A Toxic sample with a human reference and model predictions using standard language modeling loss and our proposed loss function.