# OpenReview forum: "COUNT: COntrastive UNlikelihood Text Style Transfer for Text Detoxification"
_EMNLP/2023/Conference — EMNLP 2023 Findings_

### Official Review · Reviewer_oxw4 · 2023-08-04

**Soundness:** 3

**Excitement:**

3: Ambivalent: It has merits (e.g., it reports state-of-the-art results, the idea is nice), but there are key weaknesses (e.g., it describes incremental work), and it can significantly benefit from another round of revision. However, I won't object to accepting it if my co-reviewers champion it.

**Paper Topic And Main Contributions:**

The paper proposes a new loss to fine-tune text detoxification model so it follows detoxification objectives more precisely. For instance, for text detoxification, we should control more that the output is indeed non-toxic. The paper describes the new loss function and the experiments of the model comparison trained with the loss with the baselines.

**Questions For The Authors:**

Question A: Why did not the models trained achieve the highest BLEU score?

Question B: In Table 1, there are two lines -- ParaDetox and Original LM loss. What is the difference in their meanings? ParaDetox == Bart trained on ParaDetox, then what is Original LM loss? The same for Table 2, what is the difference between BART and Original LM Loss.

Questions C: How is DiffuDetox named in these Tables?

Question D: In which cases do you think the model trained with your loss still failed? What is the further direction of improvements? Which types of toxicity cases (if it is possible to estimate) it perform better?There is a lack of discussion of the content detoxification part in the paper.

**Reasons To Accept:**

- new loss and training objective for text detoxification task;
- nice experiments design;
- the results of automatic evaluation show the overcome of model with proposed loss training over the previous baselines;

**Reasons To Reject:**

In terms of experimental setup: there is no human evaluation of the results. For ParaDetox, one of the baseline dataset, in the original paper, the automatic and human evaluation results differ in terms of the order of numbers. The paper would benefit significant with manual evaluation.

The continuation of this issue is the lack of discussion where the models are still failing. As we are trying to get the output as close as it is possible to human references, we should have BLEU metric as high as possible as well. But it is not happening in the results. I will continue the discussion of this issue in the questions section.

In the end, there are several flow in results presentation. (see questions and comments)

**Reproducibility:**

3: Could reproduce the results with some difficulty. The settings of parameters are underspecified or subjectively determined; the training/evaluation data are not widely available.

**Reviewer Confidence:**

4: Quite sure. I tried to check the important points carefully. It's unlikely, though conceivable, that I missed something that should affect my ratings.

**Typos Grammar Style And Presentation Improvements:**

Table 1 and 2: model names.

Table 5: The 2nd and the 6th examples are the same.

In experimental setup: it is nice to place the model that you were using for fine-tuning here, not in the results section. Also, please, provide the link to the specific version of the models used.

---

> ### Author Rebuttal · Authors · 2023-08-29
>
> > Human evaluation
>
> We fully agree with the reviewer that this would provide an additional useful window on the analysis.  While we do not have the financial budget to perform a comprehensive multi-annotator comparative human evaluation of all results, we note that we currently present models outputs for human inspection in Table 5 of Appendix B and we can further augment this with a limited human evaluation on a subset of examples in the Appendix (or main paper, space permitting) to determine how well the automatic evaluation correlates with human evaluation if the reviewer believes such an additional evaluation would be helpful.
>
> > Lack of discussion where the models are still failing
>
> While space in the main section of the short paper format is limited, we do present a variety of metrics in the current results to provide a multi-criteria perspective of the evaluation and we also present example outputs for human inspection in Table 5 of Appendix B.  However, we absolutely agree that more failure analysis and discussion is always useful; please see the response to Question D below for additional remarks on this and related questions.
>
> > Question A: Why did not the models trained achieve the highest BLEU score?
>
> Following the standard procedure introduced by the Paradetox evaluation kit, the BLEU score is computed between the produced paraphrase and the *toxic* input. If we have an ideal paraphrase, it does not necessarily have the highest BLEU score since some tokens that were toxic are removed or changed. The same thing happens with SIM as well. A better rephrase does not necessarily have a higher SIM since toxic parts have been removed/changed.
>
> > Question B: In Table 1, there are two lines -- ParaDetox and Original LM loss. What is the difference in their meanings? The same for Table 2, what is the difference between BART and Original LM Loss.
>
> For Table 1, ParaDetox is reported in the ParaDetox paper and the Original LM Loss is our reproduction of that method using the LM loss + BART; our reproduction of this baseline actually performs slightly better than the originally reported result so we thought it best to include it and report it separately.
>
> For Table 2, similar to Table 1, BART is the result reported in the APPDIA paper for their finetuning of BART and we evaluate outputs of that model, whereas Original LM Loss is our reproduction of this method; our reproduction generally comes very close or exceeds results reported in the paper and thus, again, we thought it best to include it and report it separately.
>
> We will clarify these distinctions on revision and also rename methods to make the similarities and distinctions between methods more suggestive from the naming alone.
>
> > Question C: How is DiffuDetox named in these Tables?
>
> We will add DiffuDetox to Table 1. The J score is 0.67, which is not better than Paradetox.
>
> > Question D: In which cases do you think the model trained with your loss still failed? What is the further direction of improvements? Which types of toxicity cases (if it is possible to estimate) it perform better?  There is a lack of discussion of the content detoxification part in the paper.
>
> These are all excellent questions, though we remark that addressing them would exceed the length limitations of the short paper format so we would have to move most of this discussion and additional analysis to the Appendix.  Nonetheless, we can certainly begin to address these questions in an expanded Appendix with more results, comparisons, and discussion.  We include some preliminary thoughts on these questions below.
>
> - Failure Analysis: From the present analysis in Appendix B, we remark that generally COUNT does an excellent job of removing explicit toxic content, but sometimes only by creating semantic ambiguity.  [WARNING: offensive content from an actual example] Notably in the last example in Appendix B, COUNT has replaced "those tits" with "they", which has masked an explicitly toxic word with a now-ambiguous reference.  We view this as a subtle semantic failure that is likely not to be caught by existing automated evaluation metrics and perhaps even some humans.  While this is only one more failure, we remark that it is not infrequent.
>
> - Directions for Future Improvements: Given this previously observed failure mode of superficially masking toxicity via semantic ambiguity (which occurs not only with the COUNT loss, but with other methods), we remark that not only is it a failure, but it is often missed by evaluation metrics.  This leads to two related but very different directions for future work: (i) It would be useful to investigate whether fine-tuning a stronger LM capable of deeper semantic reasoning (e.g., GPT-3.5 Turbo) may be able to penalize the likelihood of outputs with such semantic ambiguity under the COUNT loss, thus leading to better performance.  (ii) Since we’ve also noted that existing evaluation metrics may fail to detect such cases in the first place, this also points to the need for better automated toxicity evaluation, which may be remedied by investigating whether prompting methodologies with SOTA LMs like GPT-3.5/4 can better reason about these semantically subtle failure cases in ways that existing evaluation metrics fail to detect.
>
> - Evaluation of Different Toxicity Cases: Aside from the failure mode noted above, we have generally considered subdividing toxicity test cases according to the amount of modification needed (as a surrogate measure of detoxification difficulty) that ranges across the following categories of increasing hardness: (i) simple toxic word replacement, (ii) phrase replacement starting at the toxic word, (iii) phrase replacement starting before the toxic word (i.e. this is harder for most forward-pass beam search decoding methods), and (iv) substantial rephrasing to detoxify while preserving semantic content.  We can complete our labeling of test cases according to these categories and report these more fine-grained results for all methods if the reviewer believes such distinctions of toxicity failure cases would be insightful.
>
> We will aim to present the above discussion along with additional analysis with expanded Appendix B results (including a comparison of the output of more methods from the main paper).  If there are specific discussion points or topics the reviewer would like us to add to the discussion, we would appreciate any further suggestions.
>
> > Suggested edits, model in experimental setup, links to specific versions of models used, etc.
>
> Thanks for the careful reading and all of these suggested edits, which we will make.

---

### Official Review · Reviewer_7tdq · 2023-08-04

**Soundness:** 4

**Excitement:**

3: Ambivalent: It has merits (e.g., it reports state-of-the-art results, the idea is nice), but there are key weaknesses (e.g., it describes incremental work), and it can significantly benefit from another round of revision. However, I won't object to accepting it if my co-reviewers champion it.

**Missing References:**

Léo Laugier, John Pavlopoulos, Jeffrey Sorensen, and Lucas Dixon. 2021. Civil Rephrases Of Toxic Texts With Self-Supervised Transformers. In Proceedings of the 16th Conference of the European Chapter of the Association for Computational Linguistics: Main Volume, pages 1442–1461, Online. Association for Computational Linguistics.

This paper also tried to balance content preservation and style accuracy for the detoxification task, but in an unsupervised setting. The authors introduced a loss made of conditional auto-encoding and cycle consistency.

**Paper Topic And Main Contributions:**

The paper proposes a novel training objective function, called COUNT, for the task of strongly supervised detoxification. The idea is to penalize identity mapping often observed in previous approaches based on maximizing the likelihood of generating ground-truth. COUNT maximizes the likelihood of generating the ground-truth, relative to generating the input toxic text. The paper explains how COUNT differs from a similar approach called Unlikelihood Training. Experiments quantitatively show the benefits of COUNT over previous training objectives and state-of-the-art models.

**Questions For The Authors:**

A) Are the training steps the same across all the systems compared?

**Reasons To Accept:**

- While the objective loss is not standard, the proposed trick to palliate identity mapping is interesting
- Experiences are sound: the authors compared their method on standard parallel datasets with standard automatic evaluation metrics
- The problem statement, the method, and the results are clearly exposed.


**Reasons To Reject:**

- I wish the authors illustrated their method with examples or diagrams in order to immediately visualize the difference between COUNT, UT and standard LM. The equations are self-explanatory, yet they may necessitate some time to fully grasp the insights they offer, particularly considering that this paper could ignite the development of novel objective functions for text style transfer in the future.
- In order to draw stronger conclusions, human evaluation would have been appreciated
- As COUNT is more complex to compute, a quick analysis of its complexity relative to UT and standard LM would have been appreciated.

**Reproducibility:**

4: Could mostly reproduce the results, but there may be some variation because of sample variance or minor variations in their interpretation of the protocol or method.

**Reviewer Confidence:**

5: Positive that my evaluation is correct. I read the paper very carefully and I am very familiar with related work.

---

> ### Author Rebuttal · Authors · 2023-08-29
>
> > Visual illustration of losses
>
> We remark that we are significantly constrained by the present short paper limits, but if the reviewer has suggestions for how to visualize the losses, we’d certainly be open to a diagrammatic presentation to facilitate reader understanding.
>
> That said, we do remark that the LM loss is always a base additive term for the UT and COUNT losses so the key question mainly concerns how the UT and COUNT losses differ.
>
> In lieu of a visual illustration, we have thought of a more mathematical way to illustrate key differences in the UT and COUNT loss functions, which we summarize as follows:
>
> Formally, in the UT loss of Equation (2) and the COUNT loss of Equation (4), let us assume we can cleanly factorize the likelihood of a target style transfer example $y^*$ into two components: let $p_c$ be the probability of common content between the input x (=identity output y) and the target $y^*$, $p_t$ be the probability of toxic content in y, and $p_{nt}$ be the probability of the detoxified content in $y^*$, then we can rewrite our two losses in (2) and (4) as follows:
>
> UT: argmin  -log (1-$p_c$*$p_t$) = argmax (1-$p_c$*$p_t$)
>
> COUNT: argmin -log ($p_c$*$p_{nt}$) / [$p_c$ * $p_{nt}$ + $p_c$ * $p_t$] = argmax ($p_{nt}$ / ($p_{nt}$+$p_t$))
>
> Here we can see that the UT loss penalizes both the common (non-toxic) and non-toxic content whereas the COUNT loss *cancels* any contribution of the common components $p_c$ and isolates penalization of *only* the toxic components (i.e., there is no $p_c$ component in the COUNT loss, only $p_{nt}$ and $p_t$).
>
> Again, both UT and COUNT provide a mixture of their loss with the LM loss, which is not considered in the analysis above, but which can be just considered an orthogonal loss simply intended to ensure the output achieves high likelihood under a standard language model.
>
> If the reviewer thinks this is helpful, we will integrate this additional clarifying mathematical analysis into the main paper, or the Appendix if necessitated by space limitations.
>
> > In order to draw stronger conclusions, human evaluation would have been appreciated
>
> We fully agree with the reviewer that this would provide an additional useful window on the analysis.  While we do not have the financial budget to perform a comprehensive multi-annotator comparative human evaluation of all results, we note that we currently present models outputs for human inspection in Table 5 of Appendix B and we can further augment this with a limited human evaluation on a subset of examples in the Appendix (or main paper, space permitting) to determine how well the automatic evaluation correlates with human evaluation if the reviewer believes such an additional evaluation would be helpful.
>
> > As COUNT is more complex to compute, a quick analysis of its complexity relative to UT and standard LM would have been appreciated.
>
> Thanks, we agree this would be helpful and will add the following complexity analysis discussion to the paper:
>
> $Loss_{LM}$: requires computing $P_\theta(.|.)$ only once for $P_\theta(y=y^*|x)$.
>
> $Loss_{UT}$: requires computing $P_\theta(y=x|x)$ (i.e., y is an identity copy of x), therefore the full $Loss_{total-UT}$ (including the LM component) requires computing $P_\theta(.|.)$ twice.
>
> $Loss_{COUNT}$: requires computing $P_\theta(y=y^*|x)$ and $P_\theta(y=x|x)$, therefore the full $Loss_{total-COUNT}$ also requires computing $P_\theta(.|.)$ twice since COUNT’s $P_\theta(y=y^*|x)$ computation can be reused for the LM component.
>
> Summary: The Big-O complexity of all losses is the same, but the constants differ.  The final COUNT and UT losses each have the same computational cost, which is a 2x constant multiple of the computational cost of the LM loss.
>
> > A) Are the training steps the same across all the systems compared?
>
> No, we use early stopping on the validation loss (same stopping criterion for all methods) to determine training termination.  We found this yields the best performance for each method.
>
> > Missing EACL reference
>
> Thanks for the reference; we will certainly cite and discuss this unsupervised approach.

---

### Official Review · Reviewer_yYYJ · 2023-08-05

**Soundness:** 4

**Excitement:**

4: Strong: This paper deepens the understanding of some phenomenon or lowers the barriers to an existing research direction.

**Paper Topic And Main Contributions:**

Authors present a novel loss function - COUNT - for penalizing toxic text style by modifying the unlikelihood loss used in previous works. Proposed COUNT loss objective contrasts gold rephrased non-toxic sentence. Proposed method is shown to perform better than previously proposed loss functions on two datasets - ParaDetox and APPDIA on "J Score " metric.

**Reasons To Accept:**

- Proposed method is well justified and experimental results indicate that methods improve on baselines.
- Paper makes concise and clear improvement over previous work suitable for a short paper.
- Paper is well written and easy to follow.

**Reasons To Reject:**

- Generalization of the proposed loss to other models. Only BART is finetuned as part of the experiments.
- Supplementary missing containing code is missing, would have helped verify the implementation of the proposed method.

**Reproducibility:**

2: Would be hard pressed to reproduce the results. The contribution depends on data that are simply not available outside the author's institution or consortium; not enough details are provided.

**Reviewer Confidence:**

3: Pretty sure, but there's a chance I missed something. Although I have a good feel for this area in general, I did not carefully check the paper's details, e.g., the math, experimental design, or novelty.

---

> ### Author Rebuttal · Authors · 2023-08-29
>
> > Only BART is finetuned as part of the experiments.
>
> Given the hyperparameter sweeps required for training and evaluation, our computational resources only permitted us to compare to fine-tuned variations of BART under the different loss functions proposed.  We do note that this was sufficient to match or exceed state-of-the-art results on detoxification, which we believe presents a solid contribution and achievement for a short paper, but we acknowledge that evaluation on different encoder-decoder architectures would nonetheless be informative.  We are currently working on a T5 implementation of the proposed framework and will aim to include these comparative results in the next revision.
>
> > Reproducibility
>
> Thanks for noting these concerns.  We remark that the code to reproduce all results in this paper is in a GitHub repository whose URL will be made available in a published version of this work.

---

### Meta-Review · Area_Chair_idep · 2023-09-19

**Recommendation:** 4

**Metareview:**

This paper improves on existing work on unlikelihood training for detoxification by proposing a new objective that directly contrasts the gold standard rephrasing with the identity input-to-output mapping to focus directly on the style of the text.

Reviewers generally found this work sound, interesting and clearly presented and the improvements over the baseline compelling.

Reviewers brought up some valuable criticism based on the empiricism: lack of multiple models, human evaluation and analysis of model failures. The author rebuttal addresses many of these concerns, including an analysis of the complexity of their approach among some new empirical results.

Overall, while it’s not convincing to me that detoxification is merely style transfer, the proposed objective seems quite effective at improving the baselines presented. We hope the authors include the new results and suggested changes in the next iteration of the paper.

---

### Decision · Program_Chairs · 2023-10-07

**Decision:**

Accept-Findings

**Comment:**

This paper improves on existing work on unlikelihood training for detoxification by proposing a new objective that directly contrasts the gold standard rephrasing with the identity input-to-output mapping to focus directly on the style of the text.

Reviewers generally found this work sound, interesting and clearly presented and the improvements over the baseline compelling.

Reviewers brought up some valuable criticism based on the empiricism: lack of multiple models, human evaluation and analysis of model failures. The author rebuttal addresses many of these concerns, including an analysis of the complexity of their approach among some new empirical results.

Overall, while it’s not convincing to me that detoxification is merely style transfer, the proposed objective seems quite effective at improving the baselines presented. We hope the authors include the new results and suggested changes in the next iteration of the paper.